# Block-level Stiffness Analysis of Residual Networks

## Abstract

Residual Networks (ResNets) can be interpreted as dynamic systems, which are systems whose state changes over time and can be described with ordinary differential equations (ODEs) (Haber et al., 2018; Weinan, 2017). Specifically, the dynamic systems interpretation views individual residual blocks as ODEs. Numerical techniques for solving ODEs result in an approximation; and therefore contain an error term. If an ODE is *stiff* it is likely that this error is amplified and becomes dominating in the solution calculations, which negatively affects the accuracy of the approximated solution (Burden et al., 2015). Therefore, *stiff* ODEs are often numerically unstable. In this paper we leverage the dynamic systems interpretation to perform a novel theoretical analysis of ResNets by leveraging findings and tools from numerical analysis of ODEs. Specifically, we perform block level stiffness analysis of ResNets. We find that residual blocks towards the end of ResNet models exhibit increased stiffness and that there is a statistically significant correlation between stiffness and model accuracy and loss. Based on these findings, we propose that ResNets behave as stiff numerically unstable ODEs.

## 1 Introduction

There are three theoretical interpretations of Residual Networks (ResNets): (1) unraveled ResNets, (2) unrolled iterative estimation, and (2) dynamical systems. The unravelled interpretation views ResNets as a collection of $2^n$ paths along which the input data flows, where $n$ is the number of residual blocks (Veit et al., 2016). The unrolled iterative estimation interpretation explains ResNets as iterative approximators, where the first estimate provided by the first layer and is progressively refined by subsequent layers (Greff et al., 2017). Finally, the dynamical systems view interprets ResNets as discretized dynamical systems, where ResNets are seen as ordinary differential equations (ODEs) (Haber et al., 2018; Chen et al., 2018; Lu et al., 2018). Specifically, the dynamical systems interpretation regards ResNets's residual blocks as a series of forward Euler discretizations of an initial value ODE. This connection between residual blocks and ODEs can be leveraged for novel theoretical analyses that further our understanding and interpretation of ResNets. In this paper we perform a stiffness analysis of ResNets and their residual blocks by leveraging findings from numerical analysis of ODEs.

Stiffness is an interesting property of an ODE that has important implications. If a differential equation is *stiff*, the solution to the equation will have an unpredictable error that will negatively affect the accuracy of the approximated solution (Burden et al., 2015). Therefore, *stiff* ODEs are often numerically unstable and their solutions have accuracy issues (Seinfeld et al., 1970; Shampine & Gear, 1979).

There is no rigorous definition of stiffness; however there are certain phenomena that indicate that a problem may be stiff. One way to assess stiffness of an ODE is to analyze the eigenvalues of the Jacobian of the ODE. Specifically, if the eigenvalues of the Jacobian differ greatly in magnitude (Butcher, 2008; Bui & Bui, 1979) or if a large portion of the eigenvalues have negative real parts (Burden et al., 2015), it is likely that the ODE is stiff. Unfortunately, there are no specific thresholds regarding what constitutes a high variation in magnitude of eigenvalues or high proportion of eigenvalues with negative real parts.

In this paper we investigate whether ResNets exhibit some of the characteristics that can indicate stiffness. Specifically, we focus on analyzing the eigenvalues of the Jacobian of individual residual blocks with respect to their inputs in ResNet18, ResNet34, and ResNet50. Using these eigenvalues we calculate (1) the stiffness index and (2) proportion of eigenvalues with negative real parts for each residual block and target label, where the stiffness index captures the degree of variation of the eigenvalues magnitude (Kim et al., 2021).

We find stiffness significantly varies with respect to different residual blocks. Specifically, we find that residual blocks towards the end of the network indicate increased stiffness, i.e.: they have a high stiffness index and also a high proportion of eigenvalues with negative real parts. For example, the last block in ResNet50 has a stiffness index of -35.32 and 31.96% of the eigenvalues of its Jacobian have negative real parts.

We perform a correlation analysis between stiffness and model accuracy/loss and show that they are correlated and that their correlation is statistically significant. In particular, we calculate the pearson correlation, which ranges from -1 to 1, where positive values indicate positive linear relationships and negative values indicate inverse relationships. We also compute the p-value of the correlation coefficients, where a p-value less or equal to 0.05 is considered statistically significant. For example, given ResNet18's last residual block, the stiffness index has a negative correlation with accuracy of -0.36 and the percentage of negative eigenvalues has a positive correlation with loss of 0.34. The stiffness index of the last residual block in ResNet34 has a negative correlation with accuracy of -0.20 and the percentage of negative eigenvalues has a negative correlation with accuracy of -0.38. Finally, given the last block in ResNet50, the correlation between the stiffness index and accuracy is -0.27 and the correlation between the percent of negative eigenvalues and loss is 0.25.

Based on these findings we propose that ResNets can be interpreted as not only as ODEs, but specifically as stiff ODEs, which are numerically unstable. This interpretation could be another possible explanation of why DNNs are susceptible to adversarial examples.

The rest of the paper is organized as follows. Section 2 provides related work, Section 3 provides a detailed explanation of the dynamic systems interpretation of ResNets, which is the basis of this paper. Section 4 describes our stiffness analysis, which investigates whether individual residual blocks behave as stiff ODEs. Finally, Section 5 reports the results of our analysis and Section 6 summarizes our conclusions.

## 2 RELATED WORK

There is a large body of numerical analysis literature that studies stiffness; however, in this paper we are the first ones to connect the concepts of stiffness and ResNets via the dynamical systems interpretation (Haber et al., 2018; Weinan, 2017) to propose that ResNets can be viewed as stiff ODEs. Prior related works primarily focus on the challenges of solving stiff ODEs, which are prone to yielding unreliable results due to their stiffness.

To get an overview of on topic of stiff ODEs, please refer to Seinfeld et al. (1970), who provide a review of numerical integration techniques for stiff ODEs. Additionally, (Shampine & Gear, 1979) describe the meaning of stiffness, why do stiff problems arise, how they can be recognized, and also compare the appropriateness of different solution methods. Kaps & Rentrop (1979) propose generalized Runge-Kutta methods of order four with stepsize control as a solution method for stiff ODEs. Because stiff ODEs often contain varying time scales, Engquist & Tsai (2005) propose heterogeneous multiscale methods for stiff ODEs and show promising stability and convergence results. Enright (1978) develop a matrix updating technique that aims to reduce the computational cost of matrix operations needed for solving stiff ODEs. Other works are domain specific such as (Young & Boris, 1977), which proposes a new method for solving stiff ODEs describing the chemical kinetics of reactive flow problems.

One of the most recent works that focuses on solving stiff ODEs is Kim et al. (2021), who leverage neural ODEs. Neural ODEs have been introduced by (Chen et al., 2018) as a form of new continuous depth deep learning models that parametrize ODEs with a neural network and learn the underlying dynamic system. Kim et al. (2021) study learning neural ODEs on data generated from two classical stiff systems, ROBER (Robertson, 1967) and POLLU (Verwer, 1994), which describe the dynamics of species concentrations in stiff chemical reaction systems. They propose a new derivative calcu-

lation that can be used for neural network backpropagation and output normalization techniques for learning stiff ODEs with neural ODEs.

The dynamic systems interpretation views ResNets as a discretized forward Euler method, which is known to be a numerically unstable solution to an ODE (Press et al., 1992). Li et al. (2020) leverage this fact to establish a link between numerical stability of ResNets and their adversarial robustness. They propose modifications to the ResNet architecture that effectively replace the numerically unstable forward (explicit) Euler method with backward (implicit) Euler method, which is numerically stable. Their experiments show that the modificantion improves adversarial robustness.

## 3   DYNAMICAL SYSTEMS INTERPRETATION OF RESNETS

Dynamical system is a system whose state changes with time (Arrowsmith & Place, 1990). Dynamical systems can be discrete or continuous depending on whether the time variable is discrete or continuous. Given a dynamical system defined by function $f$, time variable $t$, and state $x$, an iteration of discrete dynamical system can be described as $x_{t+1} = f(x_t)$, while a continuous system is represented as a differential equation $dx/dt = f'(x)$. Differential equations are especially useful for describing non-linear equations, e.g.: calculating the angular position of a swinging pendulum is a canonical dynamic system that can be solved via differential equations (Brin & Stuck, 2002).

Prior works (Haber et al., 2018; Weinan, 2017) have shown that ResNets can be interpreted as ordinary differential equations, where ordinary signifies that the differential equation contains a function with respect to only one (as opposed to multiple) independent variable.

The ResNet architecture consists of a convolutional input layer followed by residual blocks with identity skip connections and a fully connected output layer. The identity skip connections allow ResNets to skip a block; and therefore, find the optimal number of layers. A residual block shown in Figure 1 is the building block of ResNets. It is composed of two convolutional layers and can be defined as:

$$x_{i+1} = F(x_i, W_i) + x_i, \tag{1}$$

where $F$ is the residual module and $W_i$ are its parameters.

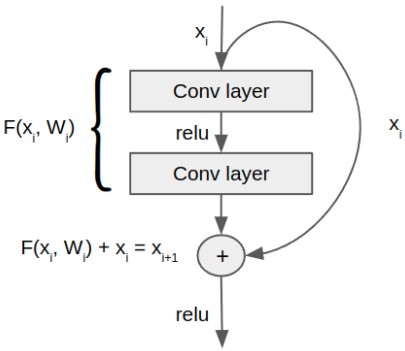

Figure 1: Residual Block

Without loss of generality, Equation 1 can be written as follows with an additional parameter $h$.

$$x_{i+1} = hF(x_i, W_i) + x_i \tag{2}$$

Equation 2 can be rearranged as:

$$\frac{x_{i+1} - x_i}{h} = F(x_i, W_i) \tag{3}$$

For a sufficiently small $h$ in Equation 3, the residual block becomes:

$$x(t) = F(x(t), W(t)), x(0) = x_0 \; for \; 0 \le t \le T, \tag{4}$$

where x(0) and x(T) correspond to the input and output feature maps and T to the depth of the residual network.

The Euler forward method is an explicit method for solving ODEs using the following formula:

$$x_{i+1} = x_i + hf(x_i, W_i), \tag{5}$$

which has the same form as Equation 2 and where $h$ represents the step size.

The equivalence of ResNets and ODENets can be demonstrated with a simple example shown in Figure 2, which compares the results of classical ResNets with ODENets that we fitted to a randomly generated set of numerical inputs and outputs.

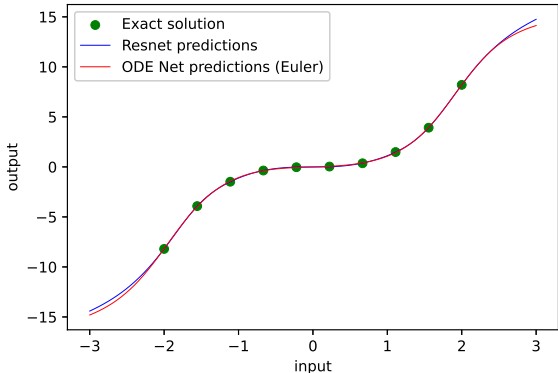

Figure 2: ResNets correspond to ODENets, which replace residual block with ODEs

## 4 STIFFNESS ANALYSIS OF RESNETS

### 4.1 STIFFNESS

Numerical solution to an ODE is an approximation; and therefore contains an error term. If an ODE is *stiff* it is likely that this error is amplified and becomes dominating in the solution calculations (Burden et al., 2015). Therefore, *stiff* ODEs are often numerically unstable. Typically, stiff ODE equations are problems for which explicit methods do not work (Hairer & Wanner, 1996; 2002).

Stiff systems derive their name from the motion of spring and mass systems that have large spring constants, and are common, for example, in the study of vibrations, chemical reactions, electrical circuits (Burden et al., 2015), and control theory (Kaps & Rentrop, 1979). There is no rigorous mathematical definition of stiffness; however certain indicators can be used to assess if an ODE is stiff. One of those indicators are the eigenvalues of the Jacobian of the function in the ODE, where the Jacobian is a matrix that consists of first order partial derivatives of a function and is defined in Equation 6. Residual blocks can be viewed as functions; and therefore, we can compute their Jacobians. Moreover, because because the Jacobians of residual blocks are square matrices, we can compute their eigenvalues.

$$\mathbb{J} = \begin{bmatrix} \dfrac{\partial \mathbf{f}(\mathbf{x})}{\partial x_1} & \cdots & \dfrac{\partial \mathbf{f}(\mathbf{x})}{\partial x_n} \end{bmatrix} = \begin{bmatrix} \nabla^T f_1(\mathbf{x}) \\ \vdots \\ \nabla^T f_m(\mathbf{x}) \end{bmatrix} = \begin{bmatrix} \dfrac{\partial f_1(\mathbf{x})}{\partial x_1} & \cdots & \dfrac{\partial f_1(\mathbf{x})}{\partial x_n} \\ \vdots & \ddots & \vdots \\ \dfrac{\partial f_m(\mathbf{x})}{\partial x_1} & \cdots & \dfrac{\partial f_m(\mathbf{x})}{\partial x_n} \end{bmatrix} \tag{6}$$

Eigenvalues are characteristic scalar values, which have special properties and can be computed only for square matrices via eigenvalue decomposition. Given a square matrix $A$, if there is a vector $X \in \mathbb{R}^n \neq 0$ such that

$$AX = \lambda X \tag{7}$$

for some scalar $\lambda$, then $\lambda$ is the eigenvalue of $A$ (Layton & Sussman, 2017). Eigenvalue decomposition of a square matrix $A$ yields three matrices: $Q$, $V$, and $Q^{-1}$ such that $A = QVQ^{-1}$. In this matrix factorization, $Q$ is a square matrix that contains eigenvectors in its columns, $V$ is a diagonal matrix that contains eigenvalues on its diagonal, and $Q^{-1}$ is simply a transpose of $Q$.

Stiff ODEs cause severe numerical integration problems and their solutions are typically associated with stability and accuracy problems (Seinfeld et al., 1970). Both individual and systems of ODEs can be stiff (Chapra & Canale, 1986). An example of a single stiff ODE is:

$$\frac{dy}{dt} = -1000y + 3000 - 2000e^{-t} \tag{8}$$

For an initial value $y(0) = 0$, the analytical solution is:

$$y = 3 - 0.998e^{-1000t} - 2.002e^{-t} \tag{9}$$

Figure 3 compares an approximate and exact solution to the example stiff ODE in Equation 8. We approximate the solution using an explicit Euler method, but even with a step size as small as 0.0015, the approximate solution is very inaccurate and unstable.

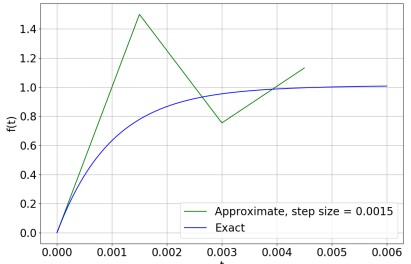 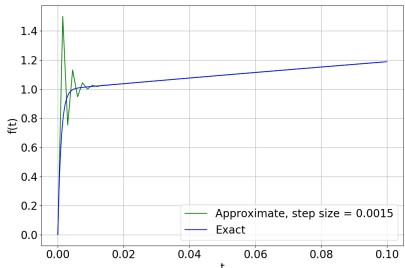

(a) Exact vs approximate solution on a time interval from 0.0 to 0.006

(b) Exact vs approximate solution on a time interval from 0.0 to 0.1

Figure 3: Solutions to stiff ODEs have accuracy and stability issues

## 4.2 STIFFNESS ANALYSIS

In section 3 we have explained how ResNets can be interpreted as a discretized forward Euler method applied to an initial value ODE as per the dynamic systems interpretation (Haber et al., 2018; Weinan, 2017). Prior works have shown that the forward Euler method is numerically unstable (Press et al., 1992) and attempted to modify ResNets to increase their stability (Li et al., 2020). Additionally, the phenomenon of adversarial examples (Goodfellow et al., 2015) that produce incorrect predictions when passed as inputs into ResNets, can be viewed as a manifestation of numerical instability (DeVore et al., 2021). Adversarial examples are inputs that have been carefully crafted with the goal of fooling a neural network. They add small virtually imperceivable perturbations to initial inputs such that the inputs appear legitimate and unaltered, but still cause the network to output an incorrect prediction. Numerically stable methods are resilient to small changes in initial data or conditions such that their output does not result in a dramatic change in the solution to the problem (Burden et al., 2015). Given these issues and the fact that the explicit methods for solving ODEs are likely to fail if a problem is *stiff*, it is possible that ResNets are stiff and numerically unstable.

To investigate this, we perform a numerical analysis to test whether ResNets exhibit characteristics of stiff ODEs. Specifically, we focus on analyzing the eigenvalues of the Jacobian of each residual block with respect to the block inputs. Using these eigenvalues, for each residual block, we calculate (1) the stiffness index and (2) what proportion of eigenvalues contain negative real parts according to equations 10 and 11 respectively.

### 4.2.1 STIFFNESS INDEX

A problem is stiff if the eigenvalues of the Jacobian of the function described by the ODE differ greatly in magnitude, which can be measured by the stiffness index (Kim et al., 2021). The stiffness

index is defined in Equation 10, where $\lambda_i$ corresponds to the eigenvalues of the Jacobian matrix and $Re$ represent the real part of a complex number.

$$S = \frac{max(Re(\lambda_i))}{min(Re(\lambda_i))} \tag{10}$$

### 4.2.2 PERCENT OF EIGENVALUES WITH NEGATIVE REAL PARTS

Another indication of stiffness is if majority of those eigenvalues have negative real parts (Heath & Munson, 1996). We calculate this metric as follows.

$$P = \frac{count(Re(\lambda_i) > 0)}{count(\lambda_i)} \tag{11}$$

### 4.2.3 COMPUTATIONAL COMPLEXITY

Computing the Jacobian of a residual block and its eigenvalues is computationally expensive. Given a residual block with input and output dimension equal to n : $F : \mathbb{R}^n \rightarrow \mathbb{R}^n$, to obtain the Jacobian matrix of $F$ we need to compute the first partial derivative of each input dimension with respect to each output dimension, which yields a n x n matrix. Therefore, the larger the inputs and residual blocks, the more more computational power is required to compute the Jacobian and its eigenvalues. We have performed experiments on GPUs as much as 80GB memory; however, that was still not sufficient for larger ResNet architectures or larger inputs. For this reason we focus on ResNet18, ResNet34, and ResNet50 that have been pre-trained on CIFAR10. Specifically, we compute the Jacobian of every residual block with respect to each input to perform the analysis. To illustrate the computational time complexity we provide the number of elements in the Jacobian matrix and the number of eigenvalues of the Jacobian for CIFAR10 ResNet18 and ResNet34 in Table 10 and 11 in the Appendix, which show that each block produces several million of these elements. For example, the first block of both ResNet18 and ResNet34 produces over 268 million Jacobian matrix elements and over 16 million eigenvalues.

## 5 RESULTS

We perform a stiffness analysis of ResNet18, ResNet34 and Resnet50 with CIFAR10 test images, and our results show that residual blocks towards the end of ResNets behave as stiff ODEs. Moreover, we show that stiffness varies with respect to different blocks as opposed to different inputs, which means that stiffness is a property of the model itself. It also implies that our results obtained using CIFAR10 images as inputs generalize across different inputs and reflect on the model as opposed to particular inputs. Additionally, we also show that there is a statistically significant correlation between elevated stiffness and model accuracy/loss.

### 5.1 RESNET18

ResNet18 contains eight residual blocks and in Table 2 we show that the stiffness index and percentage of eigenvalues with negative real parts spikes in last three blocks. The average block stiffness index is -4.15, but the last residual block shows a significant spike in stiffness of -12.99, which is three times higher than the average. The percentage of negative eigenvalues behaves similarly. The average percentage of negative eigenvalues in a block is 10.04, but this percentage doubles in the last three residual blocks. This indicates that the last three residual blocks of ResNet18 behave as stiff ODEs.

In addition to analyzing stiffness of different blocks we also calculate the average stiffness index and percentage of negative eigenvalues produced by different inputs with different target labels. In table 3 we show stiffness does not change with respect to different image labels. Based on that we conclude that it is ResNet18 itself that drives stiffness and not its inputs.

Next, we assess the significance of our finding that the last three residual blocks of ResNet18 behave as stiff ODEs by testing if the two stiffness metrics are correlated with ResNet18's accuracy and loss and if those correlations are statistically significant. In Table 1 we show that the increased stiffness in the last three blocks is correlated with ResNet18's accuracy and loss. For example, the last

| block | acc, stiffness index | | acc, pct neg eigen values | | loss, stiffness index | | loss, pct neg eigen values | |
|---|---|---|---|---|---|---|---|---|
| | corr | p-value | corr | p-value | corr | p-value | corr | p-value |
| 1 | 0.00 | 0.83 | -0.01 | 0.67 | 0.01 | 0.76 | 0.00 | 0.87 |
| 2 | 0.03 | 0.06 | -0.01 | 0.45 | -0.02 | 0.19 | 0.01 | 0.45 |
| 3 | -0.01 | 0.74 | **-0.06** | 0.00 | 0.02 | 0.17 | **0.06** | 0.00 |
| 4 | **-0.04** | 0.01 | **-0.10** | 0.00 | **0.04** | 0.03 | **0.10** | 0.00 |
| 5 | 0.00 | 0.97 | **-0.14** | 0.00 | 0.01 | 0.68 | **0.13** | 0.00 |
| 6 | **-0.09** | 0.00 | **-0.38** | 0.00 | **0.08** | 0.00 | **0.36** | 0.00 |
| 7 | -0.03 | 0.08 | **-0.18** | 0.00 | 0.02 | 0.34 | **0.18** | 0.00 |
| 8 | **-0.23** | 0.00 | **-0.36** | 0.00 | **0.21** | 0.00 | **0.34** | 0.00 |
| all blocks | -0.01 | 0.15 | **-0.07** | 0.00 | 0.01 | 0.19 | **0.07** | 0.00 |

Table 1: ResNet18 correlation analysis between stiffness and accuracy/loss with statistically significant results in bold

residual block's percentage of negative eigenvalues has a -0.36 correlation with the model accuracy and its stiffness index has a 0.34 correlation with the model loss. The p-value of these correlations is less than 0.05, i.e.: they are statistically significant. This means that higher proportion of negative eigenvalues implies lower accuracy and higher stiffness index implies higher loss.

| block | stiffness index | pct of negative eigen values |
|---|---|---|
| 1 | -2.28 | 2.82 |
| 2 | -5.12 | 2.54 |
| 3 | -1.28 | 1.96 |
| 4 | -4.04 | 4.33 |
| 5 | -1.30 | 4.35 |
| 6 | -4.94 | **19.73** |
| 7 | -1.28 | **22.65** |
| 8 | **-12.99** | **21.92** |
| all blocks | -4.15 | **10.04** |

Table 2: ResNet18 stiffness analysis by block with stiffness index and percent of negative eigen values with real parts over 10.00 in bold

| label | stiffness index | pct of negative eigen values |
|---|---|---|
| 0 | -4.31 | 10.46 |
| 1 | -4.42 | 9.71 |
| 2 | -4.02 | 10.31 |
| 3 | -4.00 | 11.57 |
| 4 | -4.02 | 9.80 |
| 5 | -4.10 | 10.14 |
| 6 | -4.07 | 9.53 |
| 7 | -4.13 | 9.20 |
| 8 | -4.15 | 9.67 |
| 9 | -4.31 | 9.99 |

Table 3: ResNet18 stiffness analysis by label

## 5.2 RESNET34

ResNet34 contains sixteen blocks and similarly to ResNet18, it is the residual blocks towards the end of the network that exhibit increased stiffness. In Table 5 we show that the last eight blocks show a spike in stiffness index and percentage of negative eigenvalues. The average block stiffness index is -7.84, but it approximately doubles in most of the last eight blocks. For example, the sixteenth block has a stiffness index of -16.82. Similar results can be observed for the percentage of negative eigenvalues in Resnet34's blocks. The average block contains 15.15 % of negative eigenvalues, but that percentage is significantly higher in the last 7 blocks. For example the last block's Jacobian matrix has 22.34 % of negative eigenvalues and the twelfth and thirteenth blocks have 33.58 % and 34.13 % respectively.

Table 6 shows that stiffness does not seem to vary with respect to different input labels, which is the same conclusion made in case of ResNet18. Compared to Table 5, which shows a block level analysis, the stiffness index and percentage of negative eigenvalues is virtually constant across different input image labels.

Similarly to ResNet18, we perform a correlation analysis to assess the impact of stiffness observed in the latter half of ResNet34's blocks on its accuracy and loss. Table 4 shows that the residual blocks with increased stiffness have a statistically significant correlation with the model's accuracy and loss. For example, the last block's stiffness index has a -0.20 correlation with the model's

| block | acc, stiffness index | | acc, pct neg eigen values | | loss, stiffness index | | loss, pct neg eigen values | |
|---|---|---|---|---|---|---|---|---|
| | corr | p-value | corr | p-value | corr | p-value | corr | p-value |
| 1 | **-0.07** | 0.02 | -0.01 | 0.74 | 0.05 | 0.09 | 0.00 | 0.94 |
| 2 | 0.02 | 0.44 | 0.01 | 0.79 | -0.03 | 0.37 | -0.01 | 0.80 |
| 3 | -0.02 | 0.42 | 0.02 | 0.56 | 0.02 | 0.53 | -0.02 | 0.54 |
| 4 | -0.01 | 0.73 | 0.03 | 0.39 | 0.00 | 0.95 | -0.02 | 0.52 |
| 5 | -0.01 | 0.82 | -0.02 | 0.48 | -0.01 | 0.82 | 0.02 | 0.59 |
| 6 | **0.06** | 0.03 | 0.00 | 0.94 | **-0.07** | 0.03 | 0.00 | 0.89 |
| 7 | 0.05 | 0.09 | **-0.12** | 0.00 | -0.04 | 0.17 | **0.12** | 0.00 |
| 8 | **-0.07** | 0.02 | **-0.22** | 0.00 | 0.04 | 0.19 | **0.23** | 0.00 |
| 9 | **-0.07** | 0.03 | **-0.26** | 0.00 | **0.06** | 0.05 | **0.26** | 0.00 |
| 10 | **-0.17** | 0.00 | **-0.32** | 0.00 | **0.17** | 0.00 | **0.31** | 0.00 |
| 11 | **-0.22** | 0.00 | **-0.31** | 0.00 | **0.21** | 0.00 | **0.30** | 0.00 |
| 12 | **-0.11** | 0.00 | **-0.23** | 0.00 | **0.10** | 0.00 | **0.22** | 0.00 |
| 13 | 0.05 | 0.08 | **-0.21** | 0.00 | -0.05 | 0.12 | **0.21** | 0.00 |
| 14 | 0.04 | 0.18 | -0.05 | 0.08 | -0.02 | 0.41 | **0.08** | 0.01 |
| 15 | **-0.23** | 0.00 | **-0.34** | 0.00 | **0.21** | 0.00 | **0.33** | 0.00 |
| 16 | **-0.20** | 0.00 | **-0.38** | 0.00 | **0.18** | 0.00 | **0.35** | 0.00 |
| all blocks | -0.01 | 0.11 | **-0.05** | 0.00 | 0.01 | 0.13 | **0.05** | 0.00 |

Table 4: ResNet34: Correlation analysis between stiffness and accuracy/loss with statistically significant results in bold

accuracy and that the percentage of negative eigenvalues in the last block have a 0.35 correlation with the model's loss.

| block | stiffness index | pct of negative eigen values |
|---|---|---|
| 1 | -3.34 | 3.10 |
| 2 | -5.24 | 2.81 |
| 3 | -5.91 | 2.78 |
| 4 | -1.10 | 2.50 |
| 5 | -6.00 | 4.78 |
| 6 | -5.69 | 4.10 |
| 7 | -5.29 | 5.15 |
| 8 | -1.73 | 7.83 |
| 9 | **-14.56** | **11.91** |
| 10 | -9.47 | **25.78** |
| 11 | **-10.46** | **30.69** |
| 12 | **-13.05** | **33.58** |
| 13 | **-11.26** | **34.13** |
| 14 | -1.85 | **25.56** |
| 15 | **-13.59** | **25.39** |
| 16 | **-16.82** | **22.34** |
| all blocks | -7.84 | **15.15** |

Table 5: ResNet34 stiffness analysis by block with stiffness index and percent of negative eigen values with real parts over 10.00 in bold

| label | stiffness index | pct of negative eigen values |
|---|---|---|
| 0 | -7.82 | 15.94 |
| 1 | -7.77 | 14.12 |
| 2 | -7.79 | 15.69 |
| 3 | -7.88 | 17.14 |
| 4 | -7.51 | 15.31 |
| 5 | -7.98 | 15.44 |
| 6 | -7.79 | 14.58 |
| 7 | -8.00 | 14.36 |
| 8 | -7.96 | 14.49 |
| 9 | -7.84 | 14.42 |

Table 6: ResNet34 stiffness analysis by label

## 5.3 RESNET50

Due to computation constraints explained in Section 4.2.3, for ResNet50, a larger architecture, we compute the stiffness index and percentage of negative eigenvalues only for its sixteenth block, which is the last residual block. The last residual block Jacobian is the smallest from all blocks as shown in Tables 10 and 11; and is therefore the least computationally expensive. Moreover, the

results from our stiffness analysis on both ResNet18 and ResNet34 show that stiff residual blocks reside towards the end of the network architecture.

As shown in Table 8, the stiffness index of the last residual block in ResNet50 is 35.32, which is higher than any block stiffness index in ResNet18 or ResNet34. The percentage of negative eigenvalues is 31.96%, which is also high. Similarly to ResNet18 and ResNet34, Table 9 demonstrates that the stiffness indicators do not significantly vary in response to different inputs. Finally, Table 7 shows that the stiffness index is about 20% correlated with ResNet50's accuracy and loss and that the correlation is statistically significant.

| block | acc, stiffness index | | acc, pct neg eigen values | | loss, stiffness index | | loss, pct neg eigen values | |
|---|---|---|---|---|---|---|---|---|
| | corr | p-value | corr | p-value | corr | p-value | corr | p-value |
| 16 | **-0.27** | 0.00 | **0.20** | 0.00 | **0.25** | 0.00 | **-0.15** | 0.00 |

Table 7: ResNet50: Correlation analysis between stiffness and accuracy/loss with statistically significant results in bold

| block | stiffness index | pct of negative eigen values |
|---|---|---|
| 16 | -35.32 | 31.96 |

Table 8: ResNet50 stiffness analysis of last block

| label | stiffness index | pct of negative eigen values |
|---|---|---|
| 0 | -31.28 | 31.91 |
| 1 | -39.61 | 28.23 |
| 2 | -32.26 | 32.79 |
| 3 | -28.45 | 35.92 |
| 4 | -33.27 | 34.14 |
| 5 | -34.33 | 36.37 |
| 6 | -39.56 | 27.23 |
| 7 | -41.05 | 32.17 |
| 8 | -37.34 | 27.77 |
| 9 | -36.30 | 33.14 |

Table 9: ResNet50 stiffness analysis by label

# 6 CONCLUSION

In this paper we leverage the dynamic systems interpretation of ResNets that views them as ODEs whose solutions are equivalent to forward Euler discretization of an initial value ODE. We perform a novel stiffness analysis whose results lead to our proposal that ResNets can be interpreted as stiff ODEs. Solutions involving stiff ODEs have numerical problems that cause poor accuracy and stability, because the error term in the approximate solution becomes dominating. While stiffness does not have a rigorous mathematical definition, there are some analyses that can be performed to assess whether its is likely that an ODE is stiff. In this paper we focus on studying the eigenvalues of the Jacobian matrix that is created by computing the first partial derivatives of a residual block with respect to its inputs. Specifically, using the eigenvalues we compute the stiffness index and percentage of negative eigenvalues with real parts for each residual block to perform a stiffness analysis. Our results show that residual blocks towards the end of ResNets exhibit a high stiffness index and high percentage of negative eigenvalues. We also show that stiffness varies with respect to different blocks as opposed to types of inputs, which indicates that stiffness is an inherent property of some residual blocks. To assess the significance of these results we perform a correlation analysis, which indicates that stiffness has a statistically significant correlation with a model's accuracy and loss. Based on these findings we conclude that some residual blocks in ResNets (specifically the ones in the latter half of the architecture) correspond to stiff ODEs. Therefore, we propose that ResNets could viewed not only as ODEs as per the dynamic systems interpretation, but specifically as stiff ODEs, which are numerically unstable. Our stiffness analysis along with instructions for reproducing results are available at: `https://anonymous.4open.science/r/Stiffness-Analysis-ResNets-18D2/README.md`.

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

# A  APPENDIX

| block | elements in Jacobian | eigenvalues in Jacobian |
|-------|----------------------|--------------------------|
| 1 | 268,435,456 | 16,777,216 |
| 2 | 268,435,456 | 16,777,216 |
| 3 | 134,217,728 | 8,388,608 |
| 4 | 67,108,864 | 8,388,608 |
| 5 | 33,554,432 | 4,194,304 |
| 6 | 16,777,216 | 4,194,304 |
| 7 | 8,388,608 | 2,097,152 |
| 8 | 4,194,304 | 2,097,152 |
| total | 801,112,064 | 62,914,560 |

Table 10: ResNet18 number of Jacobian elements and eigenvalues

| block | elements in Jacobian | eigenvalues in Jacobian |
|-------|----------------------|--------------------------|
| 1 | 268,435,456 | 16,777,216 |
| 2 | 268,435,456 | 16,777,216 |
| 3 | 268,435,456 | 16,777,216 |
| 4 | 134,217,728 | 8,388,608 |
| 5 | 67,108,864 | 8,388,608 |
| 6 | 67,108,864 | 8,388,608 |
| 7 | 67,108,864 | 8,388,608 |
| 8 | 33,554,432 | 4,194,304 |
| 9 | 16,777,216 | 4,194,304 |
| 10 | 16,777,216 | 4,194,304 |
| 11 | 16,777,216 | 4,194,304 |
| 12 | 16,777,216 | 4,194,304 |
| 13 | 16,777,216 | 4,194,304 |
| 14 | 8,388,608 | 2,097,152 |
| 15 | 4,194,304 | 2,097,152 |
| 16 | 4,194,304 | 2,097,152 |
| total | 1,275,068,416 | 115,343,360 |

Table 11: ResNet34 number of Jacobian elements and eigenvalues

