# OpenReview forum: "Block-level Stiffness Analysis of Residual Networks"
_ICLR.cc/2023/Conference — Submitted to ICLR 2023_

### Official Review · Reviewer_bzzZ · 2022-10-23

**Confidence:** 5
**Correctness:** 2
**Technical Novelty And Significance:** 3
**Empirical Novelty And Significance:** 1
**Recommendation:** 3

**Clarity, Quality, Novelty And Reproducibility:**

The idea of using tools from ODE theory to analyze resnets in this way is, to my knowledge, novel. However, there is no explanation of the experiments performed for the correlation analysis, and therefore there is no real way to understand the claims.

**Strength And Weaknesses:**

The idea of using well-established tools from ODE theory to study neural networks is an interesting one. In general leveraging well-developed insights from other fields of study to machine learning can be very fruitful.

However, the analysis conducted in this paper is very unclear and underdescribed. For example, the correlation analysis in section 5 is not described at all. What are the points being correlated? Are the same networks being used for the different types of blocks? This whole section needs a lot more explanation. Even with that explanation, it is not clear if the results are relevant to network optimization and design.

An additional minor point is that the chosen metrics for stiffness are not motivated at all. While they are well-motivated in the ODE literature, most ML researchers and practitioners will not be familiar with these notions, and it is not clear how relevant they are. For example, here stiffness corresponds to some properties of the forward pass of the network; often, poor performance of ResNets is associated with learning/generalization, and not poorly-conditioned forward passes.

**Summary Of The Paper:**

In this work the authors take advantage of the analogy between ResNets and (neural) ODEs, and try to use analysis tools from ODEs to analyze blocks in a ResNet. In particular, they focus on the notion of "stiffness" - a set of properties which make ODEs difficult to integrate, and prone to numerical instability. They claim that the stiffness metrics are inversely correlated with loss, suggesting that "stiffer" resnets are worse than non-stiff ones.

**Summary Of The Review:**

Overall, though the idea of using ODE theory to analyze resnets is interesting, the actual experiments seem lacking in both clarity and impact. Additionally, it is not clear how stiffness affects properties of the forward pass in a way which is detrimental to network performance.

---

### Official Review · Reviewer_KUvA · 2022-10-25

**Confidence:** 3
**Clarity, Quality, Novelty And Reproducibility:** See above
**Correctness:** 2
**Technical Novelty And Significance:** 3
**Empirical Novelty And Significance:** 2
**Recommendation:** 3

**Strength And Weaknesses:**

Strength: The idea of this paper is interesting, including defining the stiffness of ResNet, computing the block-wise stiffness for different models (ResNet 18, 34, and 50), and performing correlation tests between the network stiffness and their performance.

Weaknesses: Although the idea is interesting, the current manuscript is considerately below the acceptance bar of ICLR. Like in a scientific experiment, you proposed a hyperthesis, then you need to design a well-controlled experiment to verify your conjecture. The current paper is proposing a hyperthesis. It defines stiffness and poses a conjecture that "Based on these findings we propose that ResNets can be interpreted as not only as ODEs, but specifically as stiff ODEs, which are numerically unstable. This interpretation could be another possible explanation of why DNNs are susceptible to adversarial examples". To make it a convincing argument, you need to provide evidence that this conjecture is reasonable and practically meaningful. For instance, the authors argue that ResNet is stiff ODEs which may lead to vulnerability to adversarial examples. Thus, is there a way to incorporate stiffness into the training loss and demonstrate improved robustness of neural networks/ResNet?

**Summary Of The Paper:**

This paper adopts the view of interpreting individual residual blocks in ResNet as ODEs, and introduces a block-wise stiffness concept for analyzing NN performance. In numerical analysis, stiff ODEs are often numerically unstable. And the authors conjectured that stiff ODEs can be one of the reasons that DNNs are susceptible to adversarial examples. They computed the block-wise stiffness for different ResNet models, and perform statistical tests between the stiffness and model accuracy to show mode with larger stiffness might lead to a lower accuracy.

**Summary Of The Review:**

The main contribution of the paper is the introduction of stiffness in analyzing the performance of ResNet. It's an interesting yet pre-mature idea. The authors need to either provide a way of incorporating stiffness into improving the robustness of neural networks (as the authors conjectured in the paper - stiff ode is potentially one reason that NNs are vulnerable to adversarial attacks), or demonstrate potential use case and analysis to support proposed concepts.

---

### Official Review · Reviewer_j7fb · 2022-11-01

**Confidence:** 4
**Correctness:** 3
**Technical Novelty And Significance:** 3
**Empirical Novelty And Significance:** 2
**Recommendation:** 3

**Clarity, Quality, Novelty And Reproducibility:**

***Clarity***: While the overall paper is nicely written and clear to follow, the results and analysis section could be explained more rigorously.

***Quality***: The overall quality of the paper is satisfactory.

***Novelty***: The nucleus of the work is not entirely novel, but the specific application is.

***Reproducibility***: The authors provide all their code for their analysis and results, and their method is applied to open-access dataset(s) and pretrained models.

**Strength And Weaknesses:**

***Pros***:
- (Potentially) new insight into one of the most commonly used NN architectures
- Statistical testing for verification of findings

***Concerns/Weaknesses***:
- *The overall content of the paper is rather shallow*, even though there are rather obvious questions to tackle, e.g. what effect do methods have that address well-behavedness of model Jacobians (L2/Lipschitz regularization [1]), on which the stiffness analysis is solely based on? There are different types of ResNet-blocks that improve their performance (different order of operations, BatchNorm, etc. [2]) - how do they compare in regards to the stiffness analysis employed?
- *Reproducibility across datasets and different tasks*: In the paper the results only apply to pretrained ResNets on CIFAR10. Do the findings also hold for ResNets trained on different datasets and different task modalities (i.e. regression)? Even though I see that block stiffness seems to be independent of the blocks' input, I'd lack the confidence to assume that this is the case in general (as the Jacobian is in the end a function of the input when considering nonlinear blocks)
- *The paper could provide more elaborate information regarding the exact experiment/analysis setup* (at least a short section in the appendix): To me it is not directly clear how the pearson correlation is computed with regards to the accuracy/loss of the model, and how the setup gives rise to reasonable statistics if one model (pretrained ResNet<xy>) is applied to the CIFAR10. The authors could more clearly explain if the correlation is computed across test examples etc.

[1] Finlay, Chris, et al. "Lipschitz regularized deep neural networks generalize and are adversarially robust." arXiv preprint arXiv:1808.09540 (2018).

[2] He, Kaiming, et al. "Identity mappings in deep residual networks." European conference on computer vision. Springer, Cham, 2016.

**Summary Of The Paper:**

This work focuses on the concept of stiffness of ODEs, which has also found elaborate use in the Neural ODE literature. The authors argue that ResNets exhibit stiffness on the block level, i.e. view ResNets as forward Euler discretized dynamical systems where the state changes across depth rather than time. They find that stiffness proxies computed for pretrained ResNet18/34/50 models on CIFAR-10 correlate with accuracy/loss of the model with statistical significance. That is, high stiffness correlates with low accuracy / high loss. Furthermore they find that mainly blocks at the end of the network exhibit increased stiffness.

**Summary Of The Review:**

I vote for *rejection* of the paper. The work lacks some content that is needed to verify the conclusion the authors draw, i.e. that ResNets can be viewed as stiff ODEs. To me, judging by the results of the paper, this only applies to a specifc, architecturally equal ResNet models applied to a single classification dataset. Even though I see the argument of the computational burden of the analysis for large datasets and models, *I would expect the authors to verify/transfer their finding on different datasets and task modalities to vote for an accept*. Even though I respect the computational burden of the analysis, I do not see a problem with applying smaller ResNets to lower dimensional problems in order to verify the findings.

---

### Decision · Program_Chairs · 2023-01-20

**Decision:**

Reject

**Justification For Why Not Higher Score:**

see above.

**Justification For Why Not Lower Score:**

N/A

**Metareview: Summary, Strengths And Weaknesses:**

All three reviewers clearly recommend a rejection based on factual errors and limited value of the results. The authors did not engage in a discussion.